# SimBlock Simulator Enhancement with Difficulty Level Algorithm Based on Proof-of-Work Consensus for Lightweight Blockchain

**DOI:** 10.3390/s22239057

**Published:** 2022-11-22

**Authors:** Viddi Mardiansyah, Riri Fitri Sari

**Affiliations:** Department of Electrical Engineering, Faculty of Engineering, Universitas Indonesia, Depok 16424, Indonesia

**Keywords:** blockchain, lightweight blockchain, SimBlock simulator, Proof-of-Work, difficulty level

## Abstract

Proof-of-Work (PoW) was the first blockchain consensus protocol discovered, followed by Proof-of-Stake and others. The disadvantage of the PoW is that it requires high energy consumption compared to other consensus protocols. Based on this weakness, some researchers proposed a lightweight blockchain technology, a modified blockchain that has a simplified algorithm but does not reduce the security factor. This lightweight blockchain is suitable for applications requiring data reliability but with limited computing resources, such as Internet of Things devices. This paper discussed and modified the SimBlock simulator as one of the existing blockchain simulators. It has a visualization tool to look further into the propagation transition of the block. Unfortunately, the existing PoW consensus on the SimBlock simulator is unable to pinpoint the actual hash computation method. Therefore, the hashing process in the SimBlock simulator was modified by including the difficulty level for finding the hash target. The purpose of including the difficulty level was to determine how long it takes to create a block. By knowing the time needed, a recommendation could be obtained for the most suitable difficulty level for a lightweight blockchain and its implementation with IoT devices. There are two options of approaches to the difficulty level referred to in this paper; finding the number of zeros that appear sequentially and are in front of a hash value (leading zero) and finding the number of zeros that appear arbitrarily (count zero). For example, the first difficulty level on a leading-zero quest has the same meaning as searching for a leading zero, the second level of difficulty is the search for the two leading zeros, etc. The block generation time on a blockchain network using the PoW consensus highly depends on the difficulty level. Block generation time and resource utility have been analyzed and compared with other blockchain simulators and existing networks, such as Ethereum and Bitcoin. The modified SimBlock simulator was tested in this experiment using 100–600 nodes, with the expected result of creating 100–1000 blocks. Based on the experiments, creating a block using leading zeros as the hash target for the first to fourth difficulty levels took less than 1 s, whereas when using count zeros (zero-count) as the target hash, it took less than 1 s for the first to fifteenth difficulty levels. Using leading zeros took approximately 237.4 s at difficulty level 7, while count-zero took approximately 633.8 s at difficulty level 19. The experiment was not continued at the next difficulty level because it required a longer compilation time. With the modifications made, the creation of a block on a blockchain network using the PoW consensus can be clearly seen. The difficulty level added to the simulator can also provide information for determining the difficulty level to be implemented on the lightweight blockchain.

## 1. Introduction

Blockchain technology is one of the most innovative developments and discoveries that ensures data integrity in a distributed network. It is also called an immutable ledger and holds important information, such as records and ledger entries. Blockchain technology solves financial transaction issues that are typically centralized when the government or third-party organizations control all the data for every transaction [1,2,3].

The blockchain defines trustworthy, immutable, transparent, and secure data storage that works without third-party interference. The blockchain contains a data ledger of all user transactions, secures it, and distributes it (shared by various users, and each member can check the chain validity) [4,5,6]. Each block in the blockchain network stores various information. The block hash value is one of the most important for identifying a block and is attached to the next block. It can be ensured that the existing records are immutable after the block is successfully created [7].

Blockchain technology operates by using consensus mechanisms. Consensus on the blockchain is a procedure in which the peers of a blockchain network agree about the present state of the data in the network. Once a transaction is validated, it is recorded on the blockchain. It is a key feature of the decentralized nature of the blockchain. Proof-of-work (PoW) was the first consensus mechanism created [1], followed by Proof-of-Stake (PoS), Proof-of-Authority (PoA), Practical Byzantine Fault Tolerance (PBFT), and others.

The weakness of blockchain technology, especially when using the PoW consensus, is that it consumes large amounts of energy and requires high computing power. Each node miner (especially the miners for completing the predetermined target hash to create a block) requires high energy consumption to log and broadcast the general ledger information in real time. When a new node joins the network, it obtains the most updated general ledger information from the blockchain network and uses it as a reference for making the next transaction [8,9]. All the node miners compete to find the target hash to generate a block. They use a large amount of power to ensure zero downtime and create immutable data that are stored on the blockchain. Other miners waste the use of electricity and time once one of the node miners finds the target hash [8].

Currently, researchers are trying to develop a lightweight blockchain as an answer to the weakness of blockchain technology that uses the PoW consensus. A lightweight blockchain is a customized blockchain that has a simplified algorithm but does not reduce the security factor [10]. This lightweight blockchain is suitable for applications requiring data reliability but with limited computing resources, such as Internet of Things (IoT) devices.

Even though blockchain technology evolves quickly, only several blockchain simulators are available, whether it is simulating an application of a blockchain or a lightweight blockchain. A simulator is needed to show how the blockchain system works or for developers to test their applications before they are published. Faria et al. [11] proposed the BlockSim as a blockchain simulator. This simulator is a framework for simulating and modeling blockchain technology but only simulates Bitcoin and Ethereum models. Another blockchain simulator introduced by Gervais et al. [12] was developed using NS3. This simulator assesses the sturdiness of a 51% attack. Aoki et al. [13] proposed the SimBlock simulator written in Java. We further discuss this simulator in this research (see Appendix A). The SimBlock simulator has considerably more advantages than the other two simulators mentioned previously. In the SimBlock simulator, the users can easily change the behavior of nodes to enable investigation of the influence of the node behavior on blockchains. The user can easily define the mining process of a block in the SimBlock simulator. Here, the user can determine how many mining processes occur (mining processes occur per minute) to create a block in the blockchain series [13,14]. However, the working principle of the PoW in blockchain consensus cannot be identified in the SimBlock simulator.

The main contributions of this work are as follows:We enhanced the SimBlock simulator by modifying several functions and procedures in the block mining process using difficulty levels based on the PoW consensus. The difficulty level discussed in this paper is finding zeros that appear sequentially at the beginning (leading zero) or randomly (count zero) in a hash value. For example, the first difficulty level means finding a leading zero or an arbitrary number, the second difficulty level is finding two consecutive zeros in front of or in any position, and so on. These difficulty levels are used as a mathematical problem that miners must solve as proof of the PoW consensus. The block generation time on a blockchain network using the PoW consensus is highly dependent on the level of difficulty, so the duration needed to generate a block is significantly different depending on the difficulty level. The experimental results revealed the time needed to create a block based on a certain level of difficulty. These results were then analyzed to determine which level of difficulty is the most suitable for lightweight blockchain implementations.We modified the SimBlock simulator to discover the extent to which this simulator can give us recommendations on the mining process using the PoW consensus, which requires a leading-zero or count-zero search as the target hash. Therefore, in the future, we can propose new mining methods, a new hash target search, or a new consensus algorithm that is most suitable for implementing blockchain technology using IoT devices, considering the computational limitations of these devices.

The remainder of this paper is organized into several sections. Section 2 contains literature reviews of blockchain technology, lightweight blockchain, Proof-of-Work consensus, and the SimBlock simulator. In Section 3, the proposed system model is presented. The experimental setup of this research is discussed in Section 4, and the experiment results are discussed in Section 5. Finally, we provide our conclusion in Section 6.

## 2. Literature Review

### 2.1. Blockchain Technology

Blockchain technology is a process of documenting transactions on a new distributed ledger using peer-to-peer electronic transactions proposed by Satoshi Nakamoto and an immutable ledger to store the data. The stored data in a blockchain might be Bitcoin for payment history [1,15], a contract [16,17], or personal data [18].

Transactions that occur within the blockchain network are decentralized, and there is no interference from the authorities. All connected networks can see transactions that happen; therefore, the transactions are transparent [19].

All nodes in the blockchain network propagate and verify every user transaction that occurs and is successfully created in a block (generated by the node miners) through an agreed consensus mechanism [20,21]. Each node that is connected to a blockchain network records the transactions that occur within a block and between blocks connected through a mechanism of mathematical problems that are difficult to solve in a short time. The process of solving these mathematical problems is called the mining process. Nodes that carry out the mining process are called miners. A miner is a node, but not all nodes in the blockchain network are miners. Miners that can complete the mining process have the right to add a block to a blockchain network. After all the nodes verify it, the miner obtains a reward. After the block is verified and connected, the block is distributed to all the blockchain network nodes [22].

Figure 1 illustrates the standard blockchain structure, which comprises several blocks. Each block contains a header and a body. The headers in this structure commonly contain the hash values of the previous block, time-stamp, version, nonce, and Merkle root. Merkle root is the hash of all hashes of all transactions that are a part of blocks in the blockchain network. All transactions that are processed into the Merkle root are illustrated with the symbols TRX1, TRX2, TRX3, and TRX4 shown in Figure 1. These transactions originate from the blockchain user node that performs the financial transaction. For example, “Alice sends $10 to John” is TRX1, “John sends $5 to Richard” is TRX2, and so on. A Merkle tree is a structure used to efficiently verify data integrity. All transactions stored in a block are hashed to be combined with the hash value of other transactions. This hash concatenation process is repeated until there are no more hash values to be combined. The last hash is stored as a Merkle root value. The body part of the blockchain structure contains transaction data stored in the blockchain.

The implementation of blockchain technology that implements the PoW consensus has limitations. High-performance computing to process complex and extensive calculations and energy-wasteful factors is an obstacle. A race condition between nodes to pursue the desired target hash requires each node to have high-performance computing to be the first to find the target hash. This leads to wasted energy on nodes that lose out. Some researchers have proposed an alternative known as lightweight blockchain due to these constraints.

### 2.2. Lightweight Blockchain

Lightweight blockchain is currently emerging as an alternative that answers the weaknesses of blockchain technology, especially when using the PoW consensus. A lightweight blockchain is a customized blockchain with a simplified algorithm that still does not reduce the security factor when implemented [23,24]. A lightweight blockchain is suitable for applications requiring data reliability with only limited computing resources, such as IoT devices.

Previous researchers have carried out several studies on lightweight blockchains, mostly related to the technology implemented by IoT devices. Usually, the use of a lightweight blockchain is proposed due to the high-speed data transfer feature and requirement of good data security; therefore, it is challenging to change the data.

Cho et al. [25] proposed two significant points that must be fulfilled in applying blockchain technology, especially when using IoT devices. The first point is about security. Blockchain technology is currently very secure with its cryptographic techniques. However, the use of the PoW consensus in manufacturing a blockchain requires enormous computing resources, so we need a lightweight blockchain that prioritizes data security and reliability. The second point is about the quality of being lightweight. IoT devices have limitations in data storage and computing. Due to these limitations, the implementation of a blockchain on IoT devices must be able to overcome these limitations.

Dittmann et al. [26] mentioned that using a blockchain with a security feature on IoT devices could cause overload. The third-party proxy architecture was proposed to reduce the overload. Conceptually, all transactions made are sent to a proxy server so that they do not burden the IoT devices used.

Tuli et al. [24] proposed a lightweight blockchain framework that can facilitate end-to-end integrated relationships using cloud services from the IoT edge. One of the advantages of using this framework is the multi-platform facilities that provide convenience in software development.

Security is a very challenging issue in implementing IoT applications. Therefore, Bilami et al. [27] proposed a lightweight design based on a private blockchain to secure wireless communication at the device domain level. Once each node is registered for communication, it is suggested for mutual security authentication between the end device and base stations.

In this paper, the term lightweight blockchain focuses on implementing the search for the target hash value by using difficulty levels in creating a block. Modifying the SimBlock simulator by adding a difficulty levels function that applies the PoW consensus is expected to provide accurate information about the determination of the difficulty levels that are suitable for implementing IoT devices. IoT devices are known to have limited computing capacity. With a blockchain, the hashing process carried out using IoT devices has a low hash rate. For example, the Raspberry Pi 4 Model B+ generates around 90–110 hashes per second, while using a computer or laptop with an Intel i7 processor can produce around 1000–1500 hashes per second.

### 2.3. Proof-of-Work Consensus

The execution of consensus algorithms in blockchains and lightweight blockchains is indispensable and becomes a requirement that must be fulfilled before implementation because a consensus algorithm is a mechanism used by all nodes to agree on the addition of new data to the blockchain network. Proof-of-Work (PoW) is the first blockchain consensus set of rules. A blockchain uses a consensus algorithm to reach a mutual agreement about the state of the ledger distributed in its network. In this way, a blockchain builds trust between nodes that do not know each other in a distributed computing environment. The consensus algorithm ensures that each new block added to the blockchain is the only data approved by all the nodes in the blockchain network.

Blockchains typically use one of the following consensus algorithms: PoW [12], Proof-of-Stake (PoS), Proof-of-Authority (PoA), or other consensus algorithms [28,29]. The SimBlock simulator discussed in this paper only supports the blockchain simulation process using PoW and PoA consensus approaches. The adjustment made to the simulator only focused on the additional difficulty level function using the PoW consensus approach. The PoW consensus is when all the nodes in the blockchain network race to solve a riddle and create new blocks with their computation power with hash-based result values to find the nonce value [28]. Figure 2 displays the miner puzzle-solving algorithm in the PoW consensus.

Figure 2 illustrates the new transaction data combined with the previous hash and nonce values. The results of this merger are continued in the next step by using a specific hashing method, which in this case is SHA-256. The block creation process is successful if the hash value obtained has a predetermined difficulty level. Suppose the hash value does not have a specified difficulty level. In that case, the nonce value is changed to a random value and then concatenated with the data transaction and the previous hash value to repeat the process. The higher the difficulty level used, the lower the possibility of finding a hash value that has the same number of leading zeros or count zeros as the difficulty level value. This causes a lot of looping to find the hash that matches the target.

The PoW system is used to implement distributed ledger transactions on a peer-to-peer basis. This system scans the transaction value using hash methods such as SHA-1 and SHA-2 to find the hash bits. This hashing function converts/maps data of arbitrary length into fixed-length data. The output of the cryptographic hash function has a specific fixed size. The output length is dependent on the cryptographic hash method used. To ensure data security, the greater the level of the computational complexity of the cryptographic hash function, the more complex the possibility of brute-force attacks. Bitcoin uses SHA-256 [30], which outputs 256 bits or a 64-character hash. Due to its fixed-size output, infinite sets of input values can yield the same hash value for a given hash function. However, producing the same hash value from two or three similar data inputs is impossible. The two or three data inputs certainly have different meanings. For example, the sentence “John sends $10 to Anna” has a different hash value from the sentence “John sends $100 to Anna” or even from the sentence “John sends $10 to Anne”. Table 1 explains the differences in the results of the hashing process using the SHA-256 cryptographic algorithm based on the similarity of the input. This algorithm efficiently verifies the data, file, and message integrity during the transaction, data identification, and even password verification [31].

An example of this mechanism can be described as follows given data transaction, the dTx, and a target value. Concatenate the transaction using a random number of a nonce. The hash process ceases if the hash target is less or equal to the defined target. If there is no match, the nonce is randomized again, and the concatenation of the transaction with a new random nonce starts again [32,33]. For example, hashProcess(dTx||Nonce) is a hypothetical hash function; the outputs are listed in Figure 3.

The final hash value of this block is recorded as the hash value of the previous/parent block on the next block creation. When the next block is successfully created, these two blocks are called to have a link/chain between them because they have the same hash value. If there is a change in the contents of the block by an unauthorized person that causes the hash value to change, then the existing link/chain is broken because of the difference in the hash value. Therefore, if anyone wants to change the contents of a block, then they have to change all the remaining connected blocks. This takes a lot of effort.

A PoW consensus mechanism can maintain a decentralized ledger of transactions. The PoW causes the nodes involved in the mining process to compete to create blocks. However, in the process, not all miner nodes reach it, and only one node can win the race in finding the hash target; therefore, it wastes energy, capital, and time for the losing mining nodes [34].

A simulator is needed to show the process that occurs during the creation of a block and the next block and how the hash value of a block is an essential key for making the next block. The blockchain simulator can provide information about all the activities that occur when building a blockchain network.

### 2.4. SimBlock Simulator

The SimBlock simulator is one of the open-source simulators developed by the Tokyo Institute of Technology. This simulator can create a node, generate the block, transmit the block to another node as an event, and be suitable for blockchain network research [13]. The SimBlock simulator also has a visualization tool. The data presented in the JSON file make it easy for users to see the information about creating, transitioning, and propagating blocks to the entire blockchain network. This simulator demonstrates to users how creating a block in a blockchain network uses the PoW or PoA consensus.

The SimBlock simulator is available for operating systems such as Windows, macOS, Ubuntu Linux, or any Unix platform that supports Java with minimum JDK version 1.8.0 and Cradle version 5.1.1. As an event-driven simulator, each node in SimBlock behaves according to the block generation and the exchange messages with other nodes, as shown in Figure 4.

After the block is generated in the first node, an inventory message or INV message is forwarded to the second node or other nodes. The INV message contains all the block inventories, and the receiver responds with the GETDATA message to the sender. Once the sender receives the GETDATA message, the first node sends the requested block transaction to the second node [35]. However, if the second node or other node receives an INV message containing inventory transactions and existing blocks, the node ignores the INV message.

The determination of block creation time in simulation is based on the success of block generation. The mining process that requires massive calculation power is not used in this simulator. However, an assumed number of mining processes per minute is used in the process of simulation. The number of nodes created can be determined as a representation of the blockchain model. Table 2 presents the parameter used in the SimBlock simulator.

The SimBlock simulator shows the propagation transition of the block on a blockchain network. It calculates the arrival time by measuring the propagation delay between the nodes and bandwidth. The message size and bandwidth between the regions are calculated to determine the transmission time.

The design system flow in the SimBlock simulator and the additional modification process of PoW methods with the level of difficulty can be seen in Figure 5. The SimBlock simulator system flow in Figure 5 shows that to create a block using the SimBlock simulator, the user must first determine the parameter values to be used, as shown in Table 2. The REGION_LIST parameter value is used to determine the number of locations for the node to be connected. The NUM_OF_NODES parameter value determines the number of nodes connected to the blockchain network. The ENDBLOCKHEIGHT parameter value determines the number of blocks created during the simulation. The mining process is carried out by entering a value in the AVG_MINING_POWER parameter and STDEV_MINING_POWER parameter. The PoW consensus cannot be seen in this case because it has the same block generation time. Therefore, it is proposed to add a mining process using difficulty level as a mathematical problem whilst creating a block so that the SimBlock simulator can display more appropriate information when generating blocks on the blockchain.

In Figure 5, the design system flow in the SimBlock simulator starts with generating the number of regions previously defined by the user. Then, the process is continued by determining the nodes in each region and creating links between created nodes. The process starts by creating and distributing the block to all the existing nodes and calculating the propagation time needed. The block-creating process continues until the ENDBLOCKHEIGHT parameter value is reached, and the simulation ceases. The SimBlock simulation output results can be accessed in the JSON file. This file can be accessed in the output folder.

In Figure 5, the modified SimBlock simulator focuses on the block creation process using the difficulty level that applies the PoW consensus. The working principle of the PoW is to prove that to solve certain mathematical problems, several experiments are performed. When the attempt cannot solve the problem, a random nonce value is set, and the answer is recalculated. The process of randomly assigning a nonce score continues until it solves the specified mathematical problem. Here, the mathematical problem is based on the level of difficulty. Block generation in the modified SimBlock simulator starts by reading the difficulty parameter value specified by the user and assigning a nonce value at random, then combining all the existing information and performing the hashing process using SHA-256. If the hash value matches the predetermined target hash value (according to the level of difficulty), then the block creation process is declared successful, and the block distribution process throughout the network is carried out. Otherwise, the nonce variable is assigned a random value again, and the hashing process is repeated again until the hash target value is achieved.

## 3. Proposed Design

The SimBlock simulator enhancement was conducted by entering some of the coding programs and required services to support these codes. Using the built-in MessageDigest feature of the Java programming language, which has a hashing function using SHA-256, we aimed to facilitate future upgrades. Changes to the existing class structure were also made by adding nonce, hash_Data, and prev_Hash fields. A nonce value is used to store a unique number that is used as a key while combining all information to reach the hash value. This nonce is randomly generated until the hash target is achieved. hash_Data is used to store the hash value of the block, and prev_Hash is used to store the hash value of the previous block.

A detailed description of the mining puzzle-solving algorithm is provided in Figure 2, described as follows. Algorithm 1 demonstrates the function of generating the block, either a genesis block or another block/next block.
**Algorithm 1:** Creating block with mining process using difficulty.1.public Block (Block parent, long time, int nonce)
2. this.id ← latest_Id;3. this.time ← time;4. this.nonce ← nonce;5. this.hash_Data ← Call HashProcess();6. if (latest_Id = 0)} Genesis Block7.  this.prevHash ← 0;8.  this.parent ← 0;9. else} Next Blocks10.  this.parent ← parent;11.  call Mining(difficulty);12.  this.prev_Hash ← parent.getHash();13. end-if
14. increment(latest_Id);
15.end-Block;


The creation of genesis blocks in the function in Algorithm 1 is only performed once in this application. When the first block is generated, a null or zero is determined for the parent field and prev_Hash field. The hash_Data field is loaded with the value of the hash called from the HashProcess() function. The block creation process is carried out by carrying out the mining process by determining the level of difficulty. Algorithm 2 demonstrates the HashProcess() function using hash function SHA-256.
**Algorithm 2:** Process hash function using SHA-256.1.public String HashProcess()2. HashProcess ← SHA-256(this.id + this.prev_Hash + this.nonce)3. return HashProcess;4.end-HashProcess

The hashing process in Algorithm 2 uses SHA-256 provided by the Java package by concatenating field id, prev_Hash, and nonce. In the future, users can easily change the hashing method, such as MDA5, SHA-256, or others. Algorithm 3 shows the mining process using difficulty that is already declared.
**Algorithm 3:** Mining process using difficulty1.public void Mining(int Difficulty)2. Algo ← “Leading-zero”|“Counting-zero”3. while Finding_Zero(this.hash_Data, String Algo) <> Difficulty4.  this.nonce ← rand();5.  this.hash_Data ← Call HashProcess();6. end-while;7.end-Mining;

The mining process in Algorithm 3 requires difficulty parameters to determine the difficulty level in the mining process. This difficulty value is compared to the number of consecutive zeros that are present or the number of zeros that appears in the hash value obtained from the hashing process. Suppose the number of leading zeros or count zeros that exist has not met the specified difficulty value. In that case, the hashing process is carried out again by changing or providing a random value instead of the nonce field. Henceforth, the hashing process uses SHA-256. The following Algorithm 4 shows the process for calculating the number of leading zeros on a string.
**Algorithm 4:** Finding-Zero counting.1.public Finding_Zero(String hash_Data, String AlgoHash)2. if (this.hash_Data = 0)3.  return 0;4. else5.  x ← length(this.hash_Data);6.  countZero ← 0;7.  if (AlgoHash = “Leading-zero”)8.   for (z = 0; z < x; z++)9.    w ← 1 << this.hash_Data[x − 1];10.    while (this.hash_Data & r) = 011.     w ← w >> 1;12.     countZero ← rnd();13.    end-while;14.   end-for;15.  else if (AlgoHash = “Counting-Zero”)16.   for (I = 0; i < x; i++)17.     while (this.hash_Data & i) = 018.     countZero ← rnd();19.     end-while;20.    end-for21.  end-if22.  return countZero;23. end-if;24.end-Finding_Zero;

Algorithm 4 is used to find and calculate the number of zeros that appear randomly or sequentially from a hash value. The number of zeros that appear sequentially or randomly depending on the specified level of difficulty. The program returns the number of zeros found sequentially or arbitrarily and returns a zero value if it finds an empty hash value.

## 4. Experimental Setup

The experimental evaluation setup was completed using these specifications: six-core processor Intel Core i7-8750H 2.2 GHz; DDR4 memory with 32 GB capacity; primary graphics card GeForce Nvidia series RTX-2070 (GDDR6-8 GB RAM) and secondary graphics card Intel HD 630; and 1 TB Solid State Drive (SSD) as a storage device. The virtual machine from Oracle VM VirtualBox was used for the development system and installed Linux operating system Ubuntu 18.04.4 LTS. Gradle and Java JDK must be installed first before extracting the SimBlock simulator.

After all the SimBlock simulator requirement processes are successfully installed, the existing programs can be changed in the Java installation folder. Determination of parameter settings, as contained in Table 2, can be made in the Settings installation folder. After compiling the application is successful, the output of the SimBlock simulator is stored in the Output installation folder as TEXT files and JSON files.

To conduct comparative experiments under different workloads, we set the same configuration parameters on both simulators and added the PoW method with some difficulty levels to the modified SimBlock simulator. The configuration parameter test for both SimBlock simulators (the original and the modified) is shown in Table 3.

The time need to execute, memory consumption, and CPU resources needed when compiling the simulator based on the configuration settings in Table 3 were observed and compared to determine the performance of both simulators. The experiment was carried out ten times on each configuration mentioned in Table 3 to obtain a good comparison result, and then we calculated the average value. For example, an experiment with 100 nodes with a target of producing 100 blocks, 300 blocks, and 600 blocks was executed 10 times each. The process was similar for experiments with the number of nodes being 300, 600, and 1000.

## 5. Result and Discussion

The experiment with the addition of the difficulty level using the PoW consensus on the modified SimBlock was compared with the original SimBlock simulator so that the following data were obtained.

### 5.1. Mining Process Result

On a blockchain network, the mining process is when the miner nodes race to each other to complete a hash target based on a certain level of difficulty in creating a block. The original SimBlock simulator mining process is determined based on the user default value in the AVG_MINING_POWER parameter. With the SimBlock simulator application results modified with the PoW process using the difficulty level factor, the higher the value of the specified difficulty level, the longer the needed mining process. The comparison of the mining process from both simulators (the original SimBlock and the modified SimBlock) can be seen in Figure 6.

In Figure 6, the average value of the mining process was calculated from the number of hashing processes carried out to find a target hash value that is determined based on the difficulty level of all blocks and compared with the time required to create all blocks. The number of hashing processes was obtained from the sum of the nonce values in each block. The higher the difficulty level specified as the target hash, the greater the nonce value obtained.

In Figure 6, the difficulty level experiment only includes the search for up to the fifth difficulty level. It is because at the time of the experiment, the compilation process in the search for the sixth difficulty level took a long time; therefore, the experiment was ceased because it was considered a long enough time that was not workable for a simulator.

#### 5.1.1. Simulator Compilation Time

Before running the SimBlock simulator, the number of nodes and number of blocks specified in Table 3 must be inserted first in the parameter setting. Changes in the number of nodes and the number of blocks require the simulator to be re-compiled. The required compilation time of the original SimBlock simulator with predetermined parameters in Table 3 can be seen in Figure 7.

In Figure 7, the number of nodes and the number of blocks generated significantly affect the compile time required by the simulator. Based on the experimental results shown in Figure 6, the modified SimBlock test to find the target hash was only carried out up to the fourth difficulty level because of the long program compilation time that was not workable for simulator applications. The required compilation time of the modified SimBlock simulator using the parameters specified in Table 3 can be seen in Figure 8.

In Figure 8, the time required to compile the simulator program using 100, 300, 600, and 1000 nodes to create 100, 300, and 600 blocks on the first and second difficulty levels was short, and there was no significant difference. On the third difficulty level, it can be seen that the average time required to compile the simulator was twice that of the second difficulty level. On the fourth difficulty level, the time required increased by more than twice that for the third one.

The experimental results described above were obtained with a leading-zero approach. Experiments with five, six, or more difficulty levels took a long time to compile due to the computational limitations of existing equipment; therefore, the experiment was ceased.

The next experiment was to use a count-zero approach, which was predicted to be more suitable for implementation on IoT devices that are used to mine a block on a blockchain network. Figure 9 shows the time needed to find the hash target using the count-zero approach.

Figure 9 shows the results of tests of the hash target search using the count-zero approach on ten different transactions. Each transaction was tested using from the first to twentieth difficulty levels. The count-zero approach is for finding several zeros that appear arbitrarily in a hash value. Figure 9a–c shows that creating a block in a blockchain network with a count-zero approach took an average of less than 1 s, and at the 20th difficulty level in Figure 9f, it took an average of 640 s. For the 21st difficulty level, the experiment was not continued due to the same reason as the hash target search using the leading-zero approach, namely the time it takes for the program to compile.

Table 4 shows the sample transaction of hash target search using a count-zero approach, along with various difficulty levels, nonce that is matched and concatenated with the existing transaction message to produce the hash value, and the hash value itself that comprises a minimum of several zeros that appear and match with the difficulty level.

#### 5.1.2. CPU Usage Rate

A comparison of the percentage of CPU usage required to run the SimBlock with and without improvement of the PoW method with difficulty level is shown in Figure 10.

In Figure 10, the percentage of CPU usage on the original SimBlock was high, almost consuming 80% of CPU resources. Meanwhile, in the modified SimBlock, the CPU usage decreased. The more nodes involved and the more blocks created, the more CPU usage dropped below 50%. From Figure 10, the results show that the modified SimBlock proved the efficiency of CPU usage below 50%.

#### 5.1.3. Memory Consumption

A comparison of memory consumption usage that is needed to compile the SimBlock with and without improvement of the PoW method using difficulty level is shown in Figure 11.

Figure 11 shows that the memory consumption rate on the original SimBlock was always below the modified SimBlock in every configuration test conducted. Based on the observations made in each experiment, the high memory consumption rate was because of the use of the difficulty level as the determination of the target hash value. The higher the specified difficulty level, the more iterations occur to find the specified target hash. In several experiments that were conducted on the first, second, and third difficulty levels, the memory consumption rate was lower than original SimBlock. Meanwhile, in the experiment of the fourth difficulty level, the memory consumption rate was higher than the original SimBlock.

### 5.2. Discussion

Based on the experiments conducted, the addition of a mining process using difficulty level as a hash target in the SimBlock simulator was successfully implemented. The purpose of adding a mining process to the SimBlock simulator using difficulty level as the hash target was to determine how long it takes to create a block on the blockchain network for each difficulty level.

The SimBlock simulator is executed by re-compiling or re-running the program based on the specific parameter settings. The number of nodes involved and the number of blocks to be created are parameters that must be set before the simulator is re-compiled. In the modified SimBlock simulator, a difficulty level parameter was added. The value of this difficulty level parameter affects the number of zeros that appear randomly or sequentially that is to be achieved as a target hash.

Figure 6 reveals the increase in the average time required for the mining process because of the use of difficulty levels. This increment was due to more attempts to obtain the desired target hash. The higher the difficulty value, the lower the chance of successfully obtaining the target hash because more or consecutive zeros must appear. The experiment was conducted by testing all difficulty levels as the desired target hash. It started with re-compiling based on the parameters determined with the first difficulty level and then recording the time required when compiling. The same steps were repeated with the second, third, fourth, and fifth difficulty levels. However, when experimenting with the fifth difficulty, re-compiling the modified SimBlock simulator took an extended time, reaching over 5 min. Therefore, the modified SimBlock simulator experiment was only carried out up to the fourth difficulty level. We did not to continue the experiment using the fifth or higher difficulty level because the computational process was high and took a long time. Therefore, the implementation of a simulator was considered not feasible because it took too long of a wait to obtain results. However, if the hardware used has high computing power, it is possible that an experiment using the fifth or higher level of difficulty level can be conducted. We considered that this experiment discovered the most suitable difficulty level for use on a lightweight blockchain. The experiment with a higher difficulty level was not carried out.

Figure 7 shows the time required when re-compiling the original SimBlock simulator program using the parameter settings specified in Table 3. Meanwhile, Figure 8 shows the time required for re-compiling or re-running the modified SimBlock simulator program using the parameter settings specified in Table 3 combined with the first difficulty level up to the fourth difficulty level. The time required to re-compile the original SimBlock program using predetermined setting parameters was shorter than for the modified SimBlock program. The simulator ran faster because in the original program SimBlock, there is no process of searching for a hash target to create a block when re-compiling the program.

A comparison of the average level of CPU usage in the SimBlock simulator application, both the original and the modified, shown in Figure 10, showed a significant difference. The original SimBlock exhibited higher CPU usage compared to the modified SimBlock. Even in a configuration using 1000 nodes to produce 100, 300, and 600 blocks, the CPU usage was twice as high compared to the modified SimBlock. The average CPU usage level was lower than the original SimBlock simulator application, and we expect that the use of difficulty levels in this simulator is suitable for lightweight blockchain implementations.

In Figure 11, the level of memory consumption on the original SimBlock was less than that of the modified SimBlock. However, for the configurations of 600 and 1000 nodes to produce 100, 300, and 600 blocks, memory consumption requirements increased with increasing nodes. We can conclude that using more nodes in the original SimBlock simulator requires more memory. In contrast to the modified SimBlock, the memory requirements decreased.

Overall, a temporary conclusion was obtained from the experimental results that the SimBlock simulator with the addition of difficulty level as implementation of the PoW consensus shows an increase in memory consumption and CPU usage. The block generation time required by using a low level of difficulty is also considered suitable to be implemented in the IoT technology [36].

## 6. Conclusions and Future Work

In summary, the proposed modified SimBlock simulator using difficulty level to implement the PoW consensus in finding a hash target was completely developed. Visualization of the mining process of a block in the blockchain network can be seen in the output of the modified SimBlock simulator application. The output file of this simulator uses a JSON file. The data presented in the JSON file make it easier for everyone to see information about creating, transitioning, and deploying blocks throughout the blockchain network. The relationship between the genesis block and the next block in the simulation can also be seen in the block structure, where each block has a hash value that is the same as the previous hash value in the next block structure. The prev_Hash field in the genesis block has no hashed value because it is the earliest parent block or the first block in the blockchain network. In comparison, the block that is the prev_Hash field containing the hash value is the block that is linked to the previous block (the parent block). The output file also shows the nonce value of a block, indicating the unique value that was successfully obtained to generate the target hash. The hash value of a block appears to have consecutive zeros in front, showing the difficulty level used in creating this block. Based on the evidence from the output generated by the modified SimBlock simulator, we can conclude that the PoW consensus in this experiment could run well and provide more detailed information.

The CPU usage when compiling or re-running the modified SimBlock was noticeably lighter than the original SimBlock. Memory usage in the original SimBlock increased because of the number of nodes used. Meanwhile, in the modified SimBlock, the trend of memory usage also appeared to be increasing but was still lower than in the original SimBlock simulator.

The lightweight blockchain approach using the PoW consensus with a low level of difficulty from this study looks suitable to secure data originating from IoT devices. The application of the PoW consensus when using a low level of difficulty can be used because in making a block, it does not take a long time in the mining process to find the desired target hash. Even with a low level of difficulty, the data generated on the blockchain network remain secure and immutable.

Although the modified SimBlock simulator demonstrates the mining process that occurs when generating a block with leading-zero and count-zero search approaches as the hash target, the SimBlock or the modified one can still be improved in a larger scope, especially for simulating the mining process using the IoT device approach. Here are some aspects that can still be improved in the future.

Consensus selection: If the PoW consensus is still applied to this simulator, then the count-zero algorithm as a target hash value search is considered the most suitable for blockchain-based IoT devices. However, it is possible to use another consensus, such as PoA. When PoA consensus is chosen, the IoT devices that have authorization are the devices that may carry out the mining process. The IoT devices that have authorization are determined by the mechanism of the system itself.

Grouping or clustering devices as a miner: The mining process carried out by an IoT device that has limited computing processes is carried out by sharing computing power (in groups or clustering) from several existing IoT devices. Therefore, to create a block, the target hash search process can be made in groups and not by competing.

## Figures and Tables

**Figure 1 sensors-22-09057-f001:**
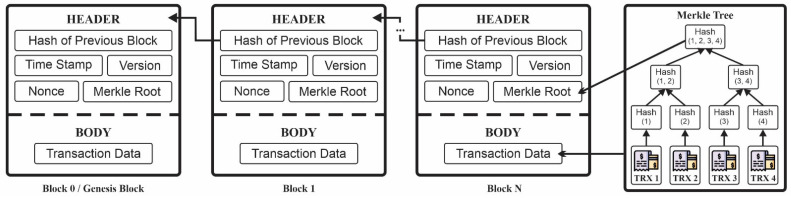
Block structure of a blockchain.

**Figure 2 sensors-22-09057-f002:**
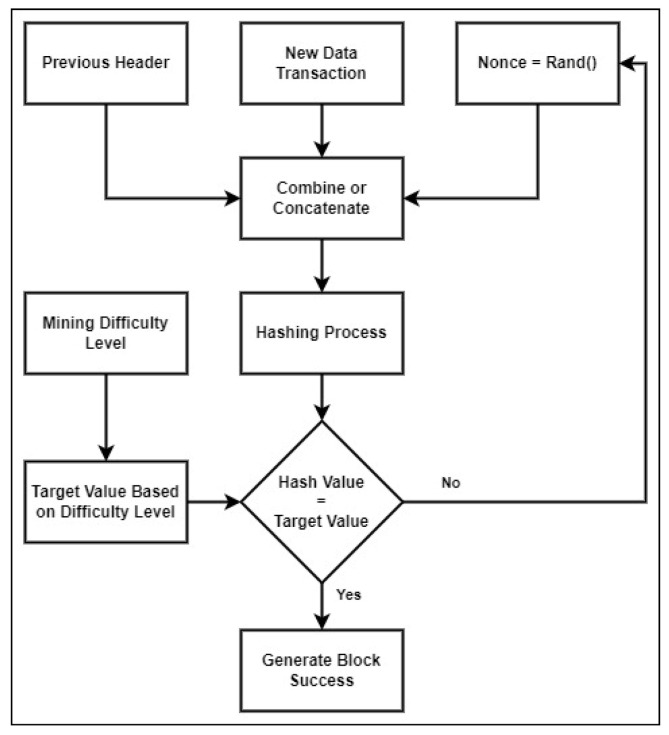
The miner puzzle-solving algorithm.

**Figure 3 sensors-22-09057-f003:**
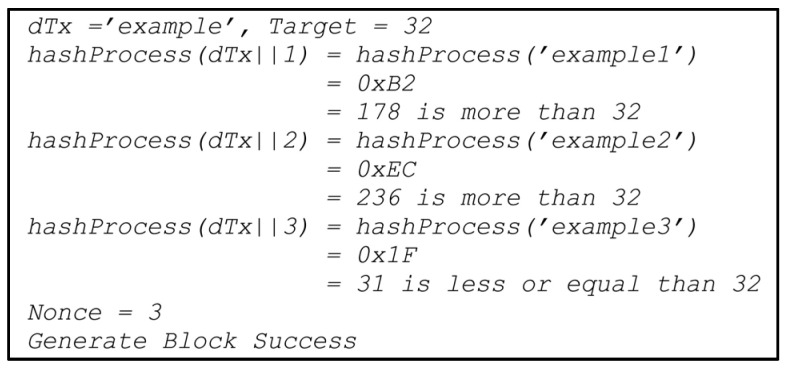
Examples of hashing processes in creating blocks on a blockchain.

**Figure 4 sensors-22-09057-f004:**
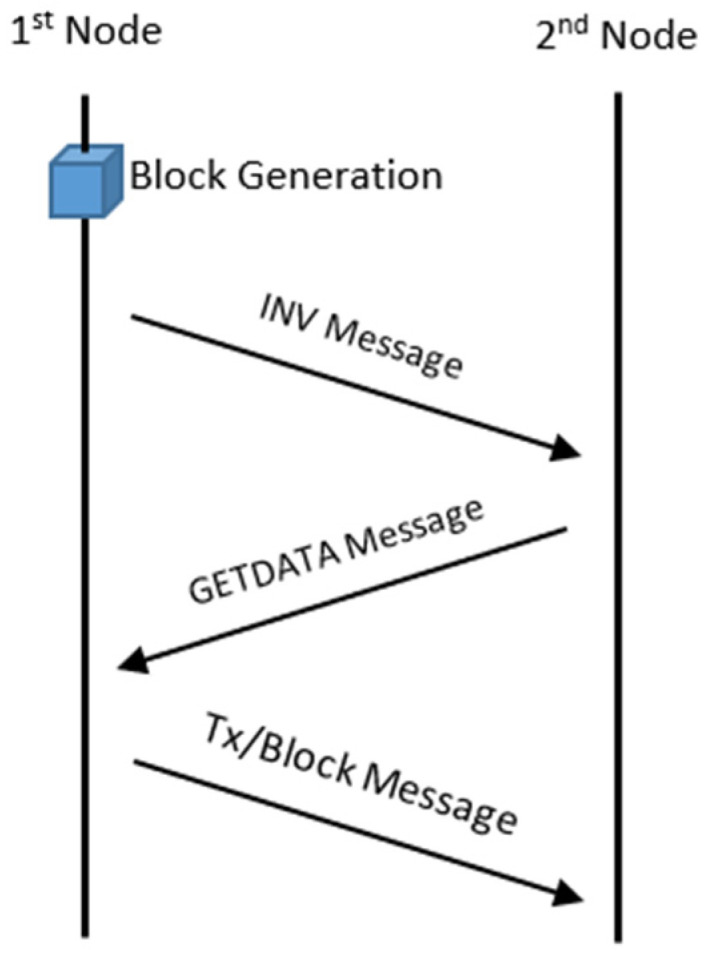
Exchanging message information between nodes.

**Figure 5 sensors-22-09057-f005:**
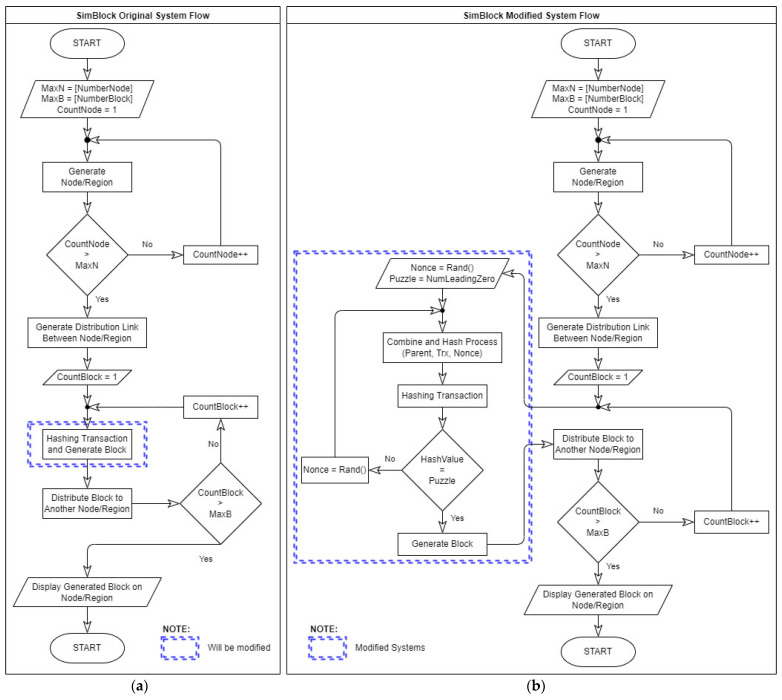
SimBlock simulator system flow (**a**) Original system; (**b**) modified system.

**Figure 6 sensors-22-09057-f006:**
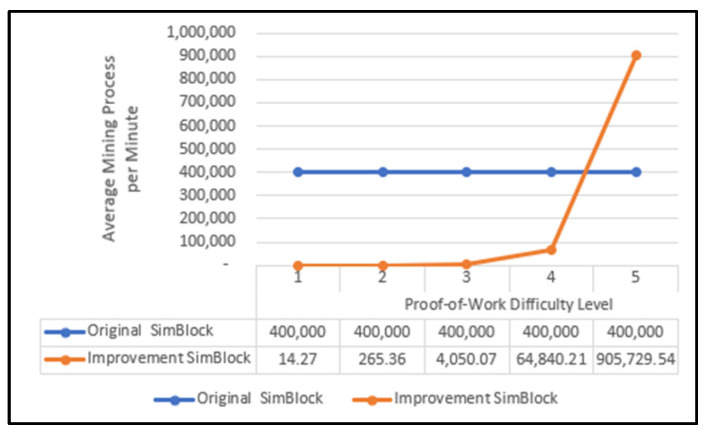
SimBlock mining process comparison.

**Figure 7 sensors-22-09057-f007:**
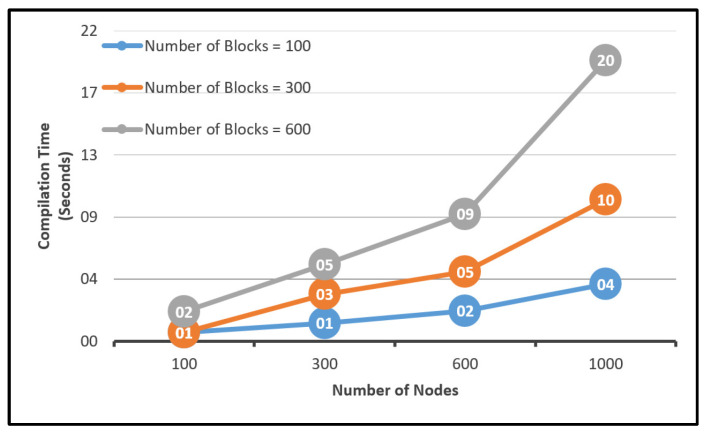
Time required to compile the original SimBlock simulator program.

**Figure 8 sensors-22-09057-f008:**
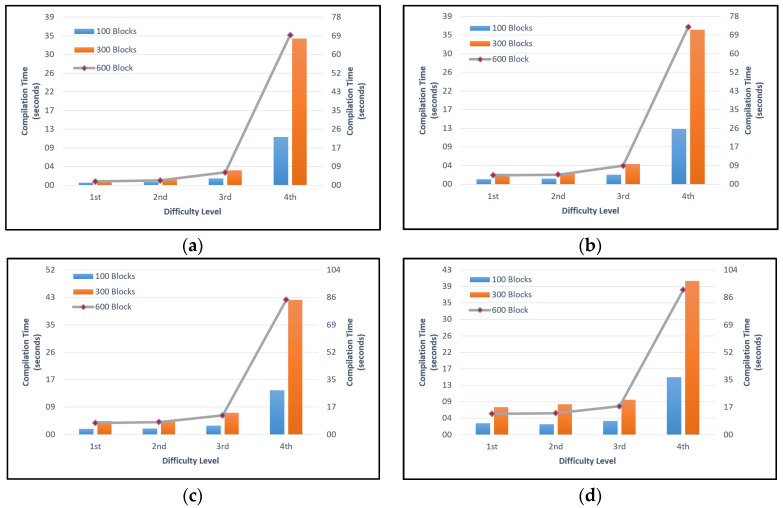
Time required to compile the modified SimBlock simulator program with (**a**) 100 Nodes; (**b**) 300 Nodes; (**c**) 600 Nodes; (**d**) 1000 Nodes.

**Figure 9 sensors-22-09057-f009:**
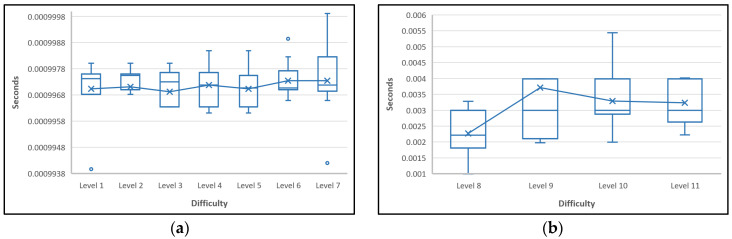
Block creation time using count-zero approach with difficulty level (**a**) 1–7; (**b**) 8–11; (**c**) 12–14; (**d**) 15–16; (**e**) 17–18; (**f**) 19–20.

**Figure 10 sensors-22-09057-f010:**
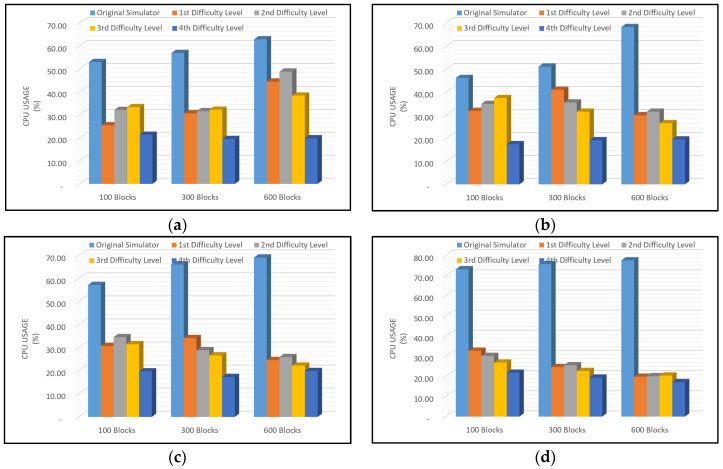
CPU Usage Rate with (**a**) 100 Nodes; (**b**) 300 Nodes; (**c**) 600 Nodes; (**d**) 1000 Nodes.

**Figure 11 sensors-22-09057-f011:**
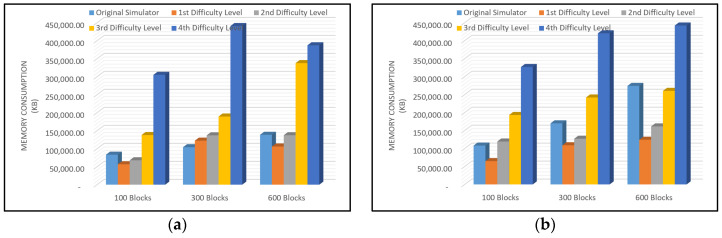
Memory Consumption Rate (**a**) 100 Nodes; (**b**) 300 Nodes; (**c**) 600 Nodes; (**d**) 1000 Nodes.

**Table 1 sensors-22-09057-t001:** Sample of the similarity message produce the different hash output when using SHA-256.

Original Message	Hashing Result Using SHA-256
John sends $10 to Anna	a2916ed42f2e4bc17801ceb0114f2020012c86972858188a0d1cebf6906cd762
**Similar Message**	**Hashing result using SHA-256**
John sends $100 to Anna	da8d307bd1453807d8f2a649cb42d49d9366a2017bbc8870bca41928675bed83
John sends $10 to Anne	529c15e8afd20b9039058ed0e80fe0a15b5a108d9417dc278860271427f09f33

**Table 2 sensors-22-09057-t002:** SimBlock simulator parameter setting.

Parameter	Variable Defined	Description
List of region	REGION_LIST	Determination of the region list.
Latency list	LATENCY	Determination of each regional latency list.
List of upload bandwidth	UPLOAD_BANDWIDTH	Determination of bandwidth upload list.
List of download bandwidth	DOWNLOAD_BANDWIDTH	Determination of bandwidth download list.
Distribution of regional	REGION_DISTRIBUTION	The distribution of the node region.
Distribution of a degree	DEGREE_DISTRIBUTION	The cumulative distribution of outbound links.
Number of nodes	NUM_OF_NODES	Nodes involved in the mining process
Routing table	TABLE	Determination of routing tables.
Block-interval	INTERVAL	The expected value of block generation interval.
Mining-power average	AVG_MINING_POWER	Average mining power of each node.
Mining-power standard deviation	STDEV_ MINING_POWER	Standard deviation of each node’s mining power.
Max block-height	ENDBLOCKHEIGHT	The maximum number of blocks.
Block size	BLOCKSIZE	Block size.

**Table 3 sensors-22-09057-t003:** Parameter setting for SimBlock simulator testing.

Number of Node	Number of Block	Number of Node	Number of Block
100	100	600	100
	300		300
	600		600
300	100	1000	100
	300		300
	600		600

**Table 4 sensors-22-09057-t004:** Sample transaction and result of count-zero approach.

Difficulty Level	Nonce Value	Sample Message	“John Sends $10 to Anne”
		Hash Value	
1	0	8160d6b9f9bb1c0b109a49bf8d742a191efe0240b4e3c661ffeb1e07f8d62dce	
2	0	8160d6b9f9bb1c0b109a49bf8d742a191efe0240b4e3c661ffeb1e07f8d62dce	
4	0	8160d6b9f9bb1c0b109a49bf8d742a191efe0240b4e3c661ffeb1e07f8d62dce	
6	0	8160d6b9f9bb1c0b109a49bf8d742a191efe0240b4e3c661ffeb1e07f8d62dce	
8	10	1840ac8cddf98e0048411c45d4b8628b8a0cfa06d0450ac0c6a5aec3b6a2f7fc	
10	80	369030f18e3085050cd1f15508c52ddff4640dfab45dcc0d98aef0033457e4ff	
12	669	0a7ee5d1004f712eea30b12970de49061e08b30001ef9517cd6ee0581e066ba3	
15	18,222	079e28ea6609cf1dd7e8c060a62ef000f64c200535806a00f0ad80a008c934e6	
18	3,723,789	a14d7105d7e5c673c7840160f0e2000a09837bf10b04030c400b080f0ed005c4	
20	379,328,593	f276b09900402d033f60909406ded04052bc1506e7f0000800963030ba3802d1	

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
