# Peer review of "SimBlock Simulator Enhancement with Difficulty Level Algorithm Based on Proof-of-Work Consensus for Lightweight Blockchain"

_sensors, 2022, doi:10.3390/s22239057_

Round 1
Reviewer 1 Report (Previous Reviewer 1)
The main aim of the article is twofold: to improve performance of SimBlock software tool and to make the tool’s blocks minting process more similar to PoW-based blockchain’s block generation process. The main article’s contribution is related to the development of block generation algorithm, which can be integrated into SimBlock code base.
Despite the article was improved in minor aspects, the major weakness wasn't addressed properly.
As it was stated earlier: it seems, that misinterpreting the blockchain simulation tasks has place, which leads to proposed algorithms inappropriateness for SimBlock’s task and misinterpretation of received experimental results. The block generating algorithm, which is currently used at SimBlock software (see relevant code lines for `Node` class `minting()` method https://github.com/dsg-titech/simblock/blob/master/simulator/src/main/java/simblock/node/Node.java#L340 and `ProofOfWork` class `minting` method https://github.com/dsg-titech/simblock/blob/master/simulator/src/main/java/simblock/node/consensus/ProofOfWork.java#L45 – both accessed 08/30/2022) is meant to simulate that each node possessing chance to produce the block. In order to reduce resources consumption instead of resource consuming hash computation the probabilistic task of random number fitting given value range is solved.
Therefore major issues:
1. The proposed solution instead of this power-friendly approach uses hash function SHA-256, which obviously consumes more computational resources, than original solution (that can be seen on fig 6). Thus proposed solution seem just to drastically reduce both possible range of cases, those can be simulated, and increase need of computational power. The real world blockchains use much more than 5 leading zeros. For instance, the recent blocks of Bitcoin has 19 leading zeros, and this can be handled by the original SimBlock solution, but this become impossible after applying authors' approach. It is unclear, where is improvement in that?
2. It is possible, that some quality metrics are sacrificed in order to improve other quality metrics, but there is no proof of that something was improved – there is only declaration, that in proposed case the simulated blockchain in some regard is more looking like the real one. But this declaration is quite disputable. There was no similarity criteria provided, which can be applied to see that this is really so.
3. There was no simulation use-case suggested, when the altered approach is better than original one. Authors don’t provide any real world example, when proposed alterations are needed. Is there any IoT lightweight blockchain that is based on PoW? What tasks authors can solve for such blockchain with the altered SimBlock?
Without addressing this aspects the proposed alterations seem to be aimless. And these are the major weaknesses those remained.
Specific style-related issues:
4. Line 227-228.”Usually, the output of the cryptographic hash function has a specific fixed size”. It’s unclear what the word “usually” is hinting at. Does there exist any case, when the cryptographic hash function has varying length output?
5. Line 230. “Blockchain uses SHA-256...” This statement need to be proved by some references.
6. Line 232-233. “However, producing the same hash value from the similarity of two or three input data is impossible.” This thesis needs to be proved or supplemented by the relevant references.
7. At the abstract and different parts of the article (lines 16-17, 77-80, 158-160) authors stated that “a lightweight blockchain technology, a modified blockchain that has a simplified algorithm but does not reduce the security factor” (this particular sentence is from lines 16-17) without referring to any work proving this property of the lightweight blockchain.
Author Response
Please see the attachment.

Reviewer 2 Report (New Reviewer)
1. In Figure 1, what are the TRX1, TRX, TRX3 and TRX4 respectively?
2. Figure 6 needs to be plotted more clearly.
3. Modifying the block-chain simulator by adding a difficulty levels function that applies the PoW consensus is expected to provide accurate information about the determination of the difficulty levels that suitable for use in the implementation of IoT devices which is generally do not have high computing power capabilitiess.
Why do not IoT devices have high computing power capabilitiess?
4. Modifying the block-chain simulator by adding a difficulty levels function that applies the PoW consensus is expected to provide accurate information about the determination of the difficulty levels that suitable for use in the implementation of IoT devices which is generally do not have high computing power capabilitiess. What does " capabilitiess " mean?
5. The final hash value in this block will be used as a parent block hash in the next block. After this value is used in the next block, this block cannot be changed without recreating the process.
Why can't this block be changed without recreating the process?
Round 2
Reviewer 1 Report (Previous Reviewer 1)
Point 1: The proposed solution instead of this power-friendly approach uses hash function SHA-256, which obviously consumes more computational resources, than original solution (that can be seen on fig 6). Thus proposed solution seem just to drastically reduce both possible range of cases, those can be simulated, and increase need of computational power. The real world blockchains use much more than 5 leading zeros. For instance, the recent blocks of Bitcoin has 19 leading zeros, and this can be handled by the original SimBlock solution, but this become impossible after applying authors' approach. It is unclear, where is improvement in that?
Response 1: Our proposal to improve SimBlock simulator capabilities is because this simulator only emphasizes output on block generation and block propagation from the blockchain network. Therefore, we propose adding a feature to see more clearly about the actual Proof-of-Work process, by taking the example of the BitCoin case. Where with bitcoin, in creating a block, the leading-zero function can show the Proof-of-Work process generated to prove that the miner node performs a raceиcondition to get the specified target hash. As well as more detailed information about a block that has been generated, can be seen properly.
The following is a comparison of the output (file output.json) from the compilation of the SimBlock simulator with the simulator we designed
Point 2: It is possible, that some quality metrics are sacrificed in order to improve other quality metrics, but there is no proof of that something was improved – there is only declaration, that in proposed case the simulated blockchain in some regard is more looking like the real one. But this declaration is quite disputable. There was no similarity criteria provided, which can be applied to see that this is really so.
Response 2: Based on the experiments we have conducted, the SimBlock simulator can simulate the blockchain creation process and the block propagation process to all nodes in the blockchain network. However, if we try to implement a simulation using the Proof-of-Work concept, which shows the leading-zero search process as in Bitcoin, then this application cannot indicate the desired results. In our research, the SimBlock simulator has stability in the hashing process, but if we want to use a leading-zero search function, we cannot see this. The SimBlock simulator that we have modified is able to show the required compile time difference due to the difficulty level required to create a block. By using leading-zero, the compile time required by our modified SimBlock simulator is indeed a long time in some experiments due to the high level of difficulty.
Point 3: There was no simulation use-case suggested, when the altered approach is better than original one. Authors don’t provide any real world example, when proposed alterations are needed.
Is there any IoT lightweight blockchain that is based on PoW? What tasks authors can solve for such blockchain with the altered SimBlock?
Without addressing this aspects the proposed alterations seem to be aimless. And these are the major weaknesses those remained.
Response 3: In this paper we have added a Github link for the coding program we have developed. This link aims to be able to show the results of our work, and can compare the final output of the SimBlock simulator that we have modified.
Here is the GitHub link: https://github.com/vmardiansyah/modified-simbloc
Answer to point 1, 2 and 3 responses:
Thank you for the explanation — it shows me the root of out misunderstanding. Indeed, your suggestion allows SimBlock to show leading-zeros utilization during the block mining process and I’m aware of this. But the reason of stating these points is to underline the scientific value. It is unclear, when one needs ‘leading zero feature’ at SimBlock. To see how Bitcoin (or any other PoW blockchain) utilize leading-zero property - it seems to be much better just to study Bitcoin blocks. You should underline the scientific value of your research, because without clear presentation of it the article may be perceived as just report on some contribution to the open source code base, which by itself is not an object of publishing at the scientific journals.
Let’s try to perceive the article from the third party point of view: your modification reduce SimBlock performance in major cases of leading zero parameter values (i.e all that are greater than 4) among tham are all of the real world cases. Moreover as I stated before, modified SimBlock cannot handle simulation of the real world case of 19 leading zeros, while the original SimBlock can. This is extremely harsh limitation — one cannot simulate any real-world blockchain. So you should state what exactly one can gain instead. What scientific task can be solved by the modified SimBlock, that cannot be solved by original SimBlock. I got it, that one is able to see additional parameters of simulation, but it remains unclear for what scientific reasons anyone want to see those parameters, how they can aid simulation process. If the curiosity is the only reason, then one can refer to the real-world blockchains, why referring to modified SimBlock? Therefore you should help readers (and me among them) to understand those reasons to make scientific novelty and practical/application value obvious.
Answer to point 4-7 responses:
Thank you for changes!
Author Response
Please see the attachment.

This manuscript is a resubmission of an earlier submission. The following is a list of the peer review reports and author responses from that submission.
Round 1
Reviewer 1 Report
The main aim of the article is twofold: to improve performance of SimBlock software tool and to make the tool’s blocks minting process more similar to PoW-based blockchain’s block generation process in order to use it in practice. The main article’s contribution is related to the development of block generation algorithm, which can be integrated into SimBlock code base.
The main strength of the article is its focus on practice and theoretic suggestions supplementation with received experimental results.
However, the article has multiple cases of inaccuracy regarding the known results interpretation, those are stated further at the review. Its main weakness is misinterpreting the blockchain simulation tasks, which leads to proposed algorithms inappropriateness for SimBlock’s task and misinterpretation of received experimental results. The block generating algorithm, which is currently used at SimBlock software (see relevant code lines for `Node` class `minting()` method https://github.com/dsg-titech/simblock/blob/master/simulator/src/main/java/simblock/node/Node.java#L340 and `ProofOfWork` class `minting` method https://github.com/dsg-titech/simblock/blob/master/simulator/src/main/java/simblock/node/consensus/ProofOfWork.java#L45 – both accessed 07/09/2022) is meant to simulate that each node possessing chance to produce the block. In order to reduce resources consumption instead of resource consuming hash computation the probabilistic task of random number fitting given value range is solved. The proposed solution solves this task in the same manner as it is done at the actual blockchains, but reducing block generation difficulty in comparison with the actual ones. However as it seems from the article that the block generation process is run for the one node and not for the whole set of simulated nodes participating at the blockchain as it is done at the original solution. If algorithm shown on fig 5 would be performed for block generation of each node all of them would get the same hash result at each iteration, therefore the first node in the set will always win the PoW contest for finding proper hash. Due to presented received performance results it’s logical to assume that authors run the algorithm just once for each iteration instead of simulating computation for each node. If the assumption is wrong it’s still unclear how authors managed to obtain compute hash(rand()) < difficultyConst quicker than rand() < difficultyConst. The performance of dTrans||nonce++ is similar to Java’s rand() in terms of big O notation (which is used for algorithm performance assessment); and in any case the time of SHA256 computation is much greater than one of rand() function. Therefore the way of authors reduce the performance time is unclear. In case I misunderstood the conception’s implementation the results should be more rigorously clarified at the article.
In any case, the scientific value of the article should be clarified more as well.
Specific comments:
1. The paragraph at the lines 55-64 is true only for blockchains those use PoW consensus, therefore this should be mentioned to avoid confusing readers.
2. Lines 60-61 contain sentence “The node miners try solving many puzzles per second to validate the block transactions.”, while actually miners doesn’t need to solve puzzles to validate transactions (those are solved at the block minting stage) they are to recompute transactions’ hash tree values in order to do so. One may refer section “8. Simplified Payment Verification” of article’s [1] in order to get more details.
3. There are excessive dots, while referring to other works (for instance, lines 64, 160)
4. At line 101 the authors statement in brackets “Every transaction that occurs (generated by the node miners) will be propagated …” is not correct. The most transactions are generated by the users, who pass them via clients software to nodes at the biggest PoW blockchains (Bitcoin, Ethereum 1.0).
5. The block structure presented on fig. 1 has several inaccuracies:
5.1 Block’s header contain more fields, than just parent block hash. It is obvious that some abstraction is in order, but ‘hash tree root’ and ‘nonce’ fields should be stated, because they are mentioned at the article, while transaction’s field nonce should be dropped to avoid readers’ misleading (indeed transactions do posses this field, but its role is irrelevant to the article’s scope and mentioning two different fields with the same title will bring only confusion to the readers).
5.2 The singular usage of “Block Transaction” may be perceived by readers as each block contains one and only one transaction, which is not correct in general case, therefore it is advised to use the plural form of the noun - “Block Transactions”.
5.3 Instead of “data hashing” it would be more proper to state “data hash value”, because hashing is a process, which cannot be integrated into data structure.
5.4 Timestamp is not a field of transaction’s body (at least it is true for the biggest PoW blockchains Bitcoin and Ethereum), therefore it is better to mention blockchain with PoW consensus, that use such block structure or drop the field or place it as part of the header.
6. The title of figure 2 is confusing. The Proof-of-Work (PoW) is a consensus conception, the term “Proof-of-Work mechanism” might be misinterpret by readers. It seems that something like “The miners’ puzzle solving algorithm” would be more proper title.
7. The very algorithm shown on figure 2 is confusing as well, in particular, the decision block should posses one input, while on figure it has two of the ones. The process “nonce = nonce + 1”, that is referred by the authors further at the article several times, (including the point I have outlined as a major weakness above) is not exactly correct. If it was so there would have been the only one winner among the miners (the one who posses the most computation power at the given time), but every miner has chance of winning the contest because they are guessing the nonce value instead of incrementing it. It’s can be difficult to generalize, because each miner can use its own tricks, but “nonce = rand(∙)” or ““nonce = nonce + rand(∙)” seem like being more proper statements to describe the process than the one used by authors.
8. The SHA-256 algorithm shouldn’t be mentioned at line 140 because SHA2 was mentioned and SHA-256 is one of the SHA2 implementations (see NIST’s FIPS 180-2 for reference).
9. The statement at lines 140-141 “This hashing crypto method creates a strong and adequate hashing code from fixed-size data to strings of character.” is not correct. Hash functions converts/maps arbitrary size data into fixed-size data. Therefore authors’ statement of fixed-size of data input data is not correct (even blockchain block’s size is varying if one strictly refers to the article topic) and the statement “strings of character” is not valid, despite data can be interpreted as one using UTF8, Unicode, Base64 etc encoding tables, but this is not a decisive feature of hash functions while “fixed-size data” is.
10. The authors’ statement at lines 142-143 “… no two input strings can produce the same 64-character hash” is not correct. Due to fixes-size output there are infinite sets of input values, those can yield to the same hash value for a given hash function. This property is the very reason why even ideal hash function is vulnerable to the preimage attack.
11. There is typo at line 144 “… event password verification”, it seems like authors meant ““… even password verification”.
12. The subsection 2.4 seems having little relation to the article, despite some common sense and wide-spread algorithm of leading zeros’ counting is mentioned further at the article. So authors might consider dropping the section. Nevertheless, the flag F, which is mentioned at 201 line, seems to be ignored in further material. Therefore, it might be better either elaborate it’s purpose/usage or not to mention it at all.
13. Presented Algorithm 1 (line 270) formally is not an algorithm – it contains pseudo-code describing data structure – there are no actions/operations. And while the Python-like pseudo-code is used the Java specific import of Java’s package at Algorithm’s line 1 seems style-inappropriate.
14. The term “lightweight blockchain”, which is stated at the title and abstract, has its first mentioning at the end of subsection 5.3 in the sentence “The lower CPU usage average rate compared to the original SimBlock simulator application is expected to be suitable for the use of Lightweight Blockchain.” and the similar statement is provided at the conclusion section, but there are neither proofs provided that proposed modifications do suit the Lightweight Blockchain nor the term “Lightweight Blockchain” explanation.
15. The paragraph at lines 381-385 contains explanation of the Algorithm 3 and doesn’t contain actual conclusions, therefore it would be proper to replace this paragraph near the Algorithm 3 description.
16. The statement at lines 393-395 regarding IoT usage wasn’t proven at the article and it is needed, because it seems infeasible due to possibility of Sybil attack from an intruder possessing regular mining equipment.
17. The statement at lines 396-398 related to security level of blockchain level wasn’t proven at the article and it seems more like it would be insecure and unstable, because the computations, those are easily simulated using laptop mentioned at lines 298-304 might be computed using proper miner equipment in time, that is far more less, than block propagation time. Therefore, it would be valid to assume that a lot of an orphan blocks would be generated and consequently problems such as double spent may arise.
Reviewer 2 Report
This paper proposes the mining process of a block in the blockchain using the PoW concept by adding several functions and procedures. It proposed the use of a leading-zero counting search as a mathematical problem that miners must solve as proof of the PoW concept,
The study covers a significant subject that falls within the series of the journal, but The paper has many weaknesses, in my opinion - in the following - I explain the highest shortcomings in terms of structure, objective, declaration of the problem.
The paper is very poorly written and requires a substantial review as well as professional proofreading services. There are many typos and the use-of-English is not up to standard. This makes it very hard to properly assess the contribution.
The abstract should be more concise, Also, it is better in the abstract to è show a specific number for the percentage of improvement
The problem/contribution of the study is not clear
What are the advantages of the proposed method over existing processes in creating blocks on a blockchain literature? These should be clearly discussed in Introduction or Related Work section
Fig. 2 is not properly discussed
"Algorithm 1: Classify the block structure " I don't know how to say that algorithm
The author should review the way of writing the algorithms In all of what was mentioned in the article to express the algorithm
In experiments How do you is the measurement / or determine the paramters no of node or no of block ?
In experiments, there are no quantitative result showing the comparison with existing